# Development and analyses of stakeholder driven conceptual models to support the implementation of ecosystem-based fisheries management in the U.S. Caribbean

Tarsila Seara[1¤]*, Stacey M. Williams[2], Kiara Acevedo[3], Graciela Garcia-Molliner[4], Orian Tzadik[5], Michelle Duval[6], Juan J. Cruz-Motta[3]

1 Department of Biology and Environmental Science, University of New Haven, West Haven, Connecticut, United States of America, 2 Institute for Socio-Ecological Research, Lajas, Puerto Rico, United States of America, 3 Department of Marine Science, University of Puerto Rico–Mayaguez, Mayaguez, Puerto Rico, United States of America, 4 Caribbean Fisheries Management Council, San Juan, Puerto Rico, United States of America, 5 National Oceanic and Atmospheric Administration, National Marine Fisheries Service, Southeast Regional Office, Aguadilla, Puerto Rico, United States of America, 6 Mellivora Consulting, West Chester, Pennsylvania, United States of America

¤ Current address: National Oceanic and Atmospheric Administration, Northeast Fisheries Science Center, Narragansett, Rhode Island, United States of America

* tseara@newhaven.edu

**Data Availability Statement:** All relevant data are within the manuscript and its Supporting Information files.

## Abstract

Fisheries management agencies in the U.S. Caribbean are currently taking steps into transitioning from a single species approach to one that includes Ecosystem Based Fisheries Management (EBFM) considerations. In this study, we developed and analyzed stakeholder-driven conceptual models with seven different stakeholder groups in Puerto Rico and the US Virgin Islands to assess and compare their perceptions of the fishery ecosystem. Conceptual models were developed for each stakeholder group during 29 separate workshops involving a total of 236 participants representing Commercial Fishers, Managers, Academics, Local Businesses, Environmental NGOs, and the Caribbean Fishery Management Council (CFMC) District Advisory Panels (DAPs) and Scientific and Statistical Committee (SSC). Non-Metric Multidimensional Scaling (nMDS) and two-mode social network analysis were used to investigate differences and similarities between stakeholder groups as well as to identify priority ecosystem elements and threats. Results show important variations between stakeholders and islands in terms of their perceived importance of ecosystem components and relationships, which supports the need for collaborative approaches and co-production of knowledge in the United States (U.S.) Caribbean region. Despite this variation, important areas of common concern among stakeholders were identified such as: habitat integrity (e.g., coral reefs), water quality, and influence of recreational fisheries and tourism on marine ecosystems. Findings of this study support the use of stakeholder-driven conceptual models as effective tools to guide decision-making, aid prioritization of data collection, and increase collaboration and cooperation among stakeholders in the context of fisheries management.

**Funding:** This work was funded by the Lenfest Ocean Program (https://www.lenfestocean.org/; 00033191), awarded to JJCM, TS, and SW. The Pew Charitable Trusts/Florida Wildlife Federation provided funds (32723) to MD to execute a portion of the data collection. The funders did not play a role in study design, data collection and analysis. However, since the manuscript was a specific deliverable required by the main funding agency, they did influence the decision to publish. The agency was not involved in the preparation of the manuscript.

**Competing interests:** The authors have declared that no competing interests exist.

# 1. Introduction

Fisheries management authorities in the U.S. have made substantial progress toward adopting ecosystem-based approaches. Under the U.S. Ecosystem-Based Fisheries Management (EBFM) Policy, NOAA Fisheries defines EBFM as "A systematic approach to fisheries management in a geographically specified area that contributes to the resilience and sustainability of the ecosystem; recognizes the physical, biological, economic, and social interactions among the affected fishery-related components of the ecosystem, including humans; and seeks to optimize benefits among a diverse set of societal goals" [1]. The focus on EBFM stems from the realization that conventional management strategies, notably single species approaches to fisheries management, have severe limitations, and generally overlook factors affecting the system at a larger scale, e.g., loss of habitat, ecological interactions, and human adaptive behavior [2, 3].

In the U.S. Caribbean, efforts to adopt an ecosystem approach to fisheries management began with a shift initiated in 2012 from a spatially undifferentiated management approach to an island-based approach, with separate stock reference points for Puerto Rico, St. Thomas/St. John (USVI), and St. Croix (USVI) (Fig 1) [4]. This new approach recognized differences in the economic, social, and cultural landscapes that shape fishing practices in the different U.S. Caribbean islands (see [4]) and establishes a basis for developing EBFM in the region.

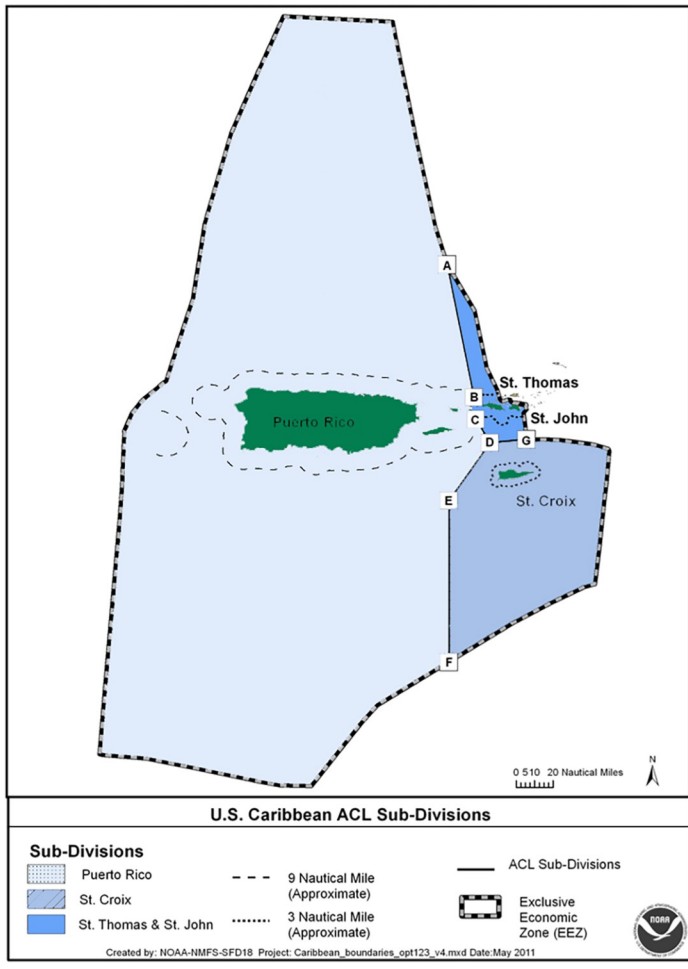

**Fig 1. Map of the U.S. Caribbean showing fishery management subdivisions (ACL subdivisions) for each island and the boundaries of the exclusive economic zone ([5]).**

Currently, the Caribbean Fishery Management Council (CFMC) EBFM Technical Advisory Panel (TAP) is tasked with developing a Fishery Ecosystem Plan (FEP) (see [6]), a document to guide the council's efforts in adopting EBFM. The process of developing the FEP in the U.S. Caribbean has been heavily based on the "FEP loop" (Fig 2) developed by the Lenfest Fishery Ecosystem Task Force (see [2]). In this loop, the process of development of an FEP is described in five stages and the cycle is repeated over time, making it adaptive. The first stage in the FEP process is characterized by the question "Where are we now?" and has the objective of guiding a model and inventory of the fishery system. The efforts described in this paper were carried out to

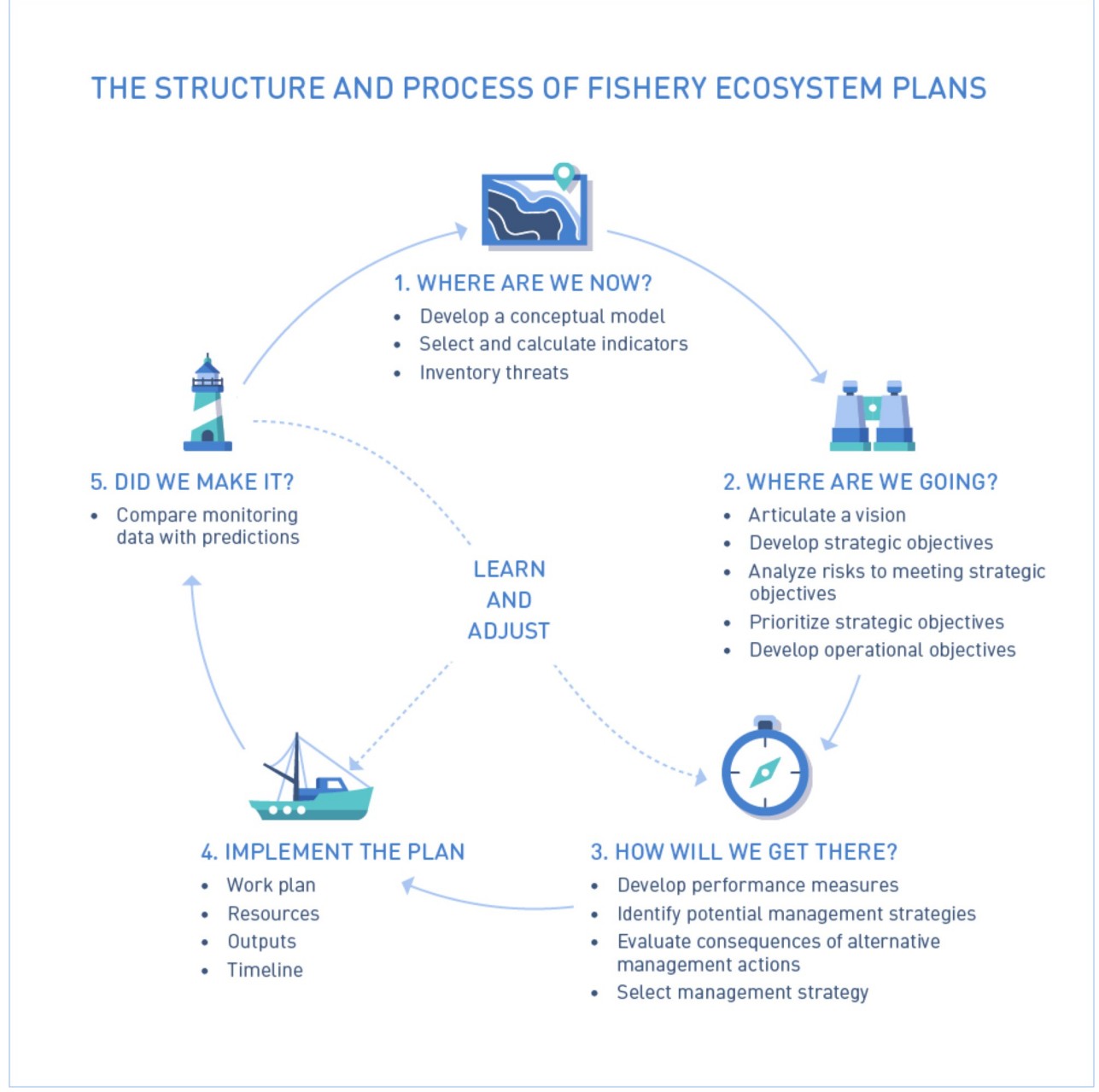

**Fig 2. FEP loop (Levin et al. 2018).**

address this initial stage, specifically through the development and analysis of conceptual models constructed with various fishery stakeholders in the U.S. Caribbean to describe the ecological, economic, social, and institutional components of the ecosystem and their causal connections.

Conceptual models are tools widely used in natural resource management. They serve various purposes, including the development of simplified schematics of complex systems, especially those with limited data [7–9]. Conceptual models also facilitate interdisciplinary collaboration [10] and provide holistic and diverse representations of social-ecological systems from the perspectives of multiple stakeholders [11, 12]. The benefits of employing conceptual models extend beyond the end product, as interactions between stakeholders during the development phase can promote and strengthen collaborative and participatory efforts. Stakeholder engagement is considered a crucial step in the successful adoption of more holistic ecosystem management strategies [3], particularly when viewed through the lens of knowledge co-production [13]. Engaging diverse stakeholders offers several advantages, including bridging the gap between traditional (e.g., scientists) and non-traditional experts (e.g. fishers), increasing stakeholders' familiarity with the decision-making process [14], and establishing a mechanism for stakeholder input, all which are associated with increased acceptance of managerial actions [3, 11, 12, 15, 16].

Developing stakeholder driven conceptual models was considered a valuable first step in the transition to EBFM in the U.S. Caribbean due to the complexity of the region's fishery ecosystem and the critical data gaps that exist in both the natural and human dimensions. Scientific knowledge gaps in the U.S. Caribbean arise from resource constraints for data collection and the management of a large number of species (over 150 stocks) within the complex tropical social-ecological systems that characterize the region [17]. This inherent complexity, including habitat interactions and diverse catch composition, results in a fishery characterized by multiple gear use and a wide range of species targeted. Other challenging characteristics of the U.S. Caribbean fisheries include the small-scale nature of the activity, the diversity of landings sites; interactions with other sectors (e.g., recreational fisheries, tourism) for which data is limited or non-existent, variations in data collection, management, enforcement between islands and jurisdictions (federal v. local), and a history of limited cooperation between agencies and stakeholders [17].

The diverse and complex characteristics of the region's fisheries, coupled with the different geographic, political, and social realities of the various islands and island groups, further highlight the limitations of single species approaches to managing the region's fisheries, and support the adoption of EBFM [17, 18]. The initial interactions with different stakeholder groups, as discussed in this paper, focused on constructing models for the entire fishery ecosystem. These models were analyzed by stakeholder group, by island/island complex, and for the U.S. Caribbean (i.e., a consensus melded model). The construction of conceptual models by different stakeholder groups in tropical socio-ecological systems, as presented here, constitutes a novel approach for fishery ecosystem level analyses. Approaching the conceptual modelling exercise from an ecosystem scale was considered a critical step to understand this complex fishery system from a stakeholder viewpoint and obtain information that can inform the prioritization of policy and management actions and data needs.

## 2. Materials and methods

### 2.1. Data collection: Development of stakeholder driven conceptual models

The conceptual models presented in this study were developed through multiple concurrent efforts led by different groups. The effort led by this study's authors, referred to as Cruz-Motta et al. in Table 1, engaged with commercial fishers ("fishers"), fisheries management personnel

**Table 1. Summary information about different workshops held to develop conceptual models with multiple U.S. Caribbean fisheries stakeholder group.**

| Stakeholder Group | Number of Models | Number of Participants | Island Grouping (Number of Workshops)* | Group Leading Workshops | Method | Year |
|---|---|---|---|---|---|---|
| Commercial Fishers** | 12 | 117 | PR (10); STT (1); STX (1) | Cruz-Motta et al. | In person | 2021/ 22 |
| Managers | 3 | 9 | PR (2); USVI (1) | Cruz-Motta et al. | Virtual | 2021 |
| Academics | 4 | 24 | PR (3); USVI (1) | Cruz-Motta et al. | Virtual | 2021 |
| Businesses | 3 | 17 | PR (1); STT (1); STX (1) | Mellivora Consulting | Virtual | 2021 |
| NGOs | 3 | 20 | PR (1); STT (1); STX (1) | Mellivora Consulting | Virtual | 2021 |
| DAPs | 3 | 38 | PR (1); STT (1); STX (1) | CFMC | In person | 2018 |
| SSC | 1 | 11 | U.S. Caribbean | CFMC | In person/ Virtual | 2018/ 19 |
| *Total* | *29* | *236* | - | - | - | - |

*PR = Puerto Rico; STT = St. Thomas, USVI; STX = St. Croix, USVI

**Commercial fishers are those engaged in fishing activities with the purpose of selling their catch in the market.

("managers"), and fishery science academics ("academics"). Mellivora Consulting led the development of conceptual models with local fishery related businesses ("businesses") and environmental Non-Governmental Organizations ("NGOs"). The CFMC was responsible for the efforts involving their District Advisory Panels ("DAPs") and Scientific and Statistical Committee ("SSC"). District Advisory Panels are comprised of individuals with diverse experience and interest in the local fishing industry who provide information and recommendations to the Council. Members include commercial and recreational fishers, local businesses, scientists, and other members of the public (see: https://caribbeanfmc.com/about-us/cfmc-district-advisory-panels). The Scientific and Statistical Committee is comprised of scientists from state and Federal agencies, academic institutions, and other affiliations. Members of the SSC review management plans and other documents to ensure the council is using the best available science in their decision-making process (see: https://caribbeanfmc.com/about-us/cfmc-scientific-and-statistical-committee). Coordination between the different efforts to develop conceptual models helped to maximize the compatibility of employed methodologies and their outcomes. Separate conceptual models were developed for each stakeholder group (see Table 1).

These stakeholders represented the islands of Puerto Rico, St. Thomas/St. John, and St. Croix, and conceptual models were developed by island/island group, except for the managers and academics models which were developed for Puerto Rico and the USVI (combining USVI islands), and the SSC model which was developed for the U.S. Caribbean (Puerto Rico and USVI combined). All 29 stakeholder workshops were held between 2018 and 2022 involving a total of 236 participants. Due to COVID-19 restrictions, workshops took place online, except for those with the DAPs and commercial fishers. The DAPs models were constructed over the course of three meetings with each group organized and led by the CFMC. Fishers' workshops were organized in 10 different communities around Puerto Rico and one each in St. Thomas and St. Croix. The SSC model was developed during multiple meetings held and organized by the CFMC which were conducted both in person and online. Participants in the online workshops were recruited by email from an initial list compiled by the researchers with the assistance of collaborators with experience working with the respective stakeholder groups. Fishers were recruited through flyers, social media posts, word of mouth, and direct communication by researchers and local fishery liaisons.

During stakeholder workshops, conceptual models were developed using a multi-step fuzzy cognitive mapping approach based on Ozesmi and Ozesmi [19]. Participants were given an

introductory brief summarizing the objectives of the conceptual models within the context of the Caribbean FEP development, and an unrelated example (e.g., a forest ecosystem) to illustrate the process of building a conceptual model by identifying components and relationships between them. Following the introduction, prompt questions were used to initiate the building of the model. Participants were first asked to identify important components of the entire U.S. Caribbean marine fishery system, which could be social, biological, economic, cultural, or physical in nature, and then prompted to link these different components based on relationships between them. These relationships could be positive or negative, meaning that components could affect each other by increasing or decreasing their quantity or quality. Some relationships were characterized as neutral when the direction of the effect could either vary depending on the conditions, or involve components with optimal levels (e.g., salinity) and thus be impossible to characterize as exclusively negative or positive.

During online workshops, models were constructed by the research team on the screen using Mental Modeler online software [20]. In all in-person workshops, models were built using post-it notes attached to a large white notepad and linked by arrows drawn by workshop moderators based on input from attendees. After each virtual workshop, participants were sent a link via email to access the model built during their workshop and had 2 weeks to provide comments and thus validate the model and increase process transparency [21]. Models constructed in person were validated that same day. Final versions of each individual model were created using Mental Modeler software and underlying information of each model was saved as matrices for further analysis. Matrix rows listed individual relationship between components (including signs +/-), and columns represented the different stakeholder groups/ islands. Matrix cells then indicated presence (1)/absence (0) of each relationship in the respective stakeholder groups' conceptual models.

Approval from the University of New Haven Institutional Review Board (Protocol 2023–051) was obtained prior to the analyses and publication of results to ensure that the use of the qualitative data initially collected with the purpose of informing management strategies conformed with the ethical standards for research involving human subjects. The complete set of data for analyses was obtained and compiled in July 2023. The manner with which the data was collected and recorded makes the identification of individuals involved in the construction of conceptual models virtually impossible, therefore consent was not obtained. The researchers have taken the appropriate steps to guarantee confidentiality of individuals upon presentation of methods and results.

## 2.2. Methods of conceptual models data analyses

**2.2.1. Development of consensus models.** All components identified by stakeholders in each conceptual model were thoroughly examined for terminology cohesiveness. Only components that were deemed identical or very similar were combined as to maintain the integrity of the conceptual models while avoiding weakening relationships due to the use of synonyms by different participants and stakeholder groups. Some idiosyncratic components were further combined into broader categories (e.g., specific types of pollution grouped under non-point source pollution). Conceptual model data were then merged to create a unique consensus model for the U.S. Caribbean using social network analysis methodology in Gephi software version 0.10.1 (Yifan Hu layout) [22]. Models by stakeholder group and by island/island group were also created. Consensus models reflect the most important fishery ecosystem components and relationships based on the collective perceptions of the participant stakeholder groups.

**2.2.2. Stakeholder group similarity analysis.** A presence/absence matrix containing all relationships between components identified in stakeholders' conceptual models and

attributed to each stakeholder group by island/island group was used to analyze similarity between groups using multivariate ordinations. Non-Metric Multidimensional Scaling (nMDS) was used to visualize patterns of similarities between stakeholder groups and islands, based on a simple matching index of identified relationships among ecosystem components. In addition, statistically significant groupings were identified using the routine SIMPROF, in which the null hypothesis of no grouping is constructed using permutations of original data. All multivariate analyses were performed using the PRIMER-e software [23].

**2.2.3 Identification of priority ecosystem components.** Matrices listing components identified as either drivers (components affecting other components) or receivers (components being affected by another component) by each stakeholder group were created to develop two-mode social network analysis diagrams using UCINET 6 [24] and NetDraw version 2.181 [25] software for the U.S. Caribbean and by island/island group. These diagrams were used to identify convergence and divergence between stakeholder groups with regard to ecosystem components identified as either drivers or receivers.

## 3. Results

### 3.1. Consensus models

The consensus model displayed in Fig 3 provides a visual representation of the extent of information obtained through the development of multiple stakeholder driven conceptual models for the U.S. Caribbean (See S1 File for island/island group-based analysis). All ecosystem components identified by stakeholders are included in Fig 3A and are linked to each other by lines representing relationships that are either negative (red), positive (green), or neutral (yellow). The size of the letters is indicative of the number of connections involving each component and the thickness of the lines represents the number of stakeholder groups that mentioned the relationship, providing a visualization of ecosystem elements and relationships that can be considered important or as priorities for participant stakeholders. Fig 3B is a sub-model of Fig 3A including only the relationships that represent the highest level of agreement between stakeholder groups, i.e., were identified by 5 (maximum) or 4 stakeholder groups, again represented by line thickness (see also Table 2). Number of connections is also emphasized in Fig 3B by the size of the font, with the most prominent components being Coral Reefs, Fisheries Resources, and Water Quality. Table 2 lists the relationships displayed in Fig 1B, and the respective stakeholder groups that included them in their conceptual models.

Stakeholders generally agreed that impacts of climate change, coastal development and pollution have negative effects on marine habitats, including coral reefs, mangroves and seagrass beds. Relationships involving law enforcement, management, and illegal fishing were also identified by multiple stakeholder groups. The relationships with the highest level of agreement among stakeholder groups are the impacts of run-off (negative), water quality (positive), and marine diseases (negative) on coral reefs, the impacts of coastal development on run-off (positive), and the effect of Education and Outreach efforts on compliance with fisheries rules and regulations (positive) (Table 2).

### 3.2. Stakeholder group similarity

Patterns of spatial distribution can be identified in Fig 4 and, despite a medium to high stress value of 0.2, can be used as indicative of relative differences and similarities between groups regarding the relationships between components identified in each group's conceptual models. These analyses suggest that, generally, similarity of perceptions regarding important relationships between ecosystem components is relatively high among similar groups of stakeholders

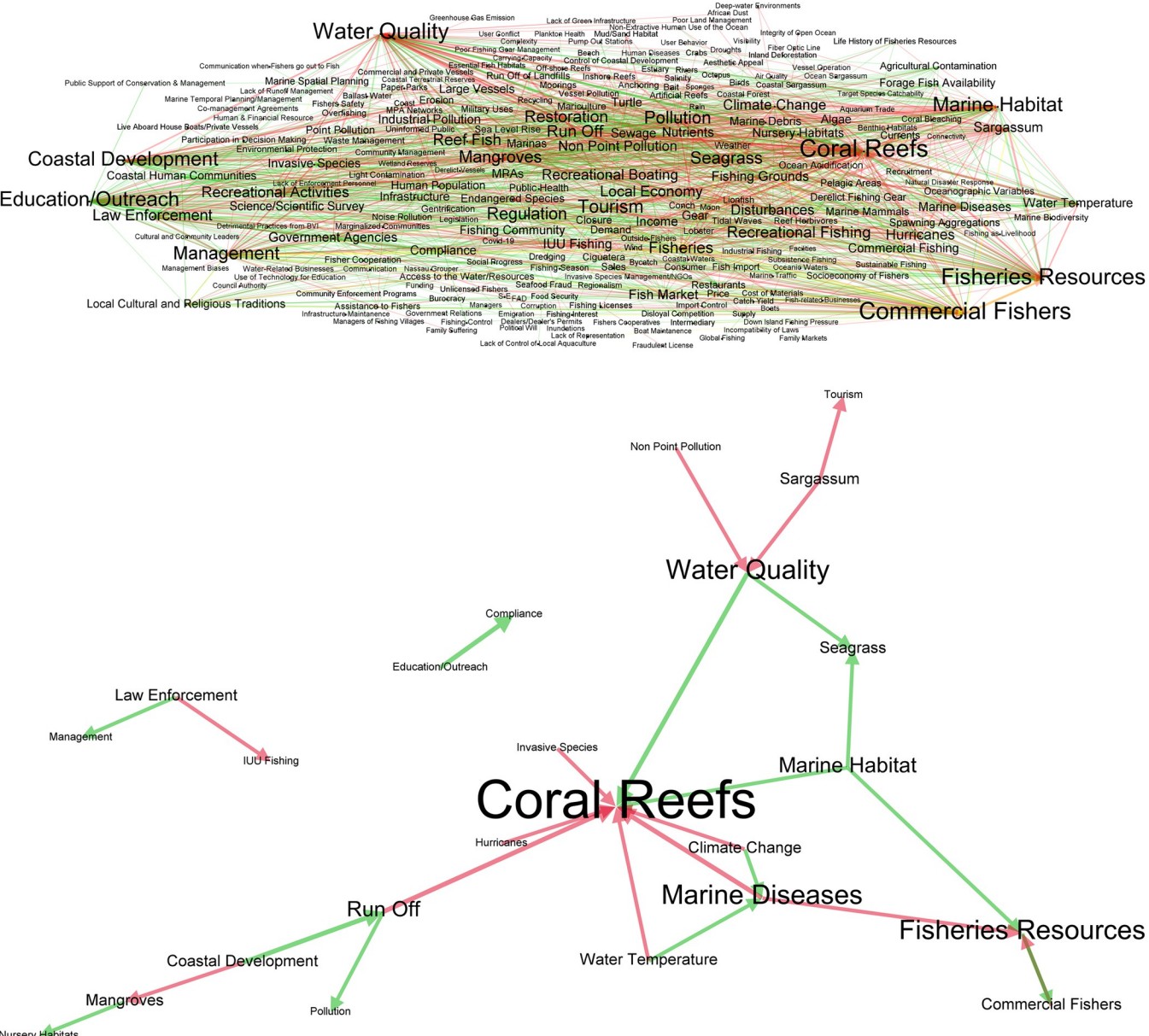

**Fig 3.** A. Consensus conceptual model for the U.S. Caribbean, merging all conceptual models for the seven stakeholder groups for all island/island groups. B. Consensus conceptual model for the U.S. Caribbean showing only relationships present in at least four stakeholder group's conceptual models. Arrow direction characterizes components as drivers (affecting other components) or receivers (being affected by other components). Red, green, and yellow lines represent negative, positive, and neutral relationships, respectively. Letter size is reflective of the frequency with which component was mentioned by stakeholders and thickness of the lines reflects agreement, i.e., the number of stakeholder groups that mentioned each relationship.

regardless of the island/island group to which they belong. The highest disparity is observed among the DAPs.

Detailed analysis of the correlation between each original variable (i.e., relationships between components) and the axis ordinations revealed that relationships involving the importance of coral reefs for fishery resources and the effect of commercial fishery activities on these resources (which is indicative of dependence and impact), were important for the stakeholder groupings on the left side of the plot, which includes fishers, managers, and the

**Table 2. List of top relationships by stakeholder agreement level and respective stakeholder groups for the U.S. Caribbean.** Number of groups refer to the total number of stakeholder groups that identified each relationship in their conceptual model(s).

| Relationship | Sign | MAN | EXP | DAPs | BUS | NGO | FISH | SSC | # of Groups |
|---|---|---|---|---|---|---|---|---|---|
| Run Off-Coral Reefs | - | 1 | 1 | 0 | 1 | 1 | 1 | 0 | 5 |
| Marine Diseases-Coral Reefs | - | 0 | 1 | 1 | 1 | 1 | 1 | 0 | 5 |
| Water Quality-Coral Reefs | + | 0 | 1 | 1 | 1 | 1 | 0 | 1 | 5 |
| Coastal Development-Run Off | + | 0 | 1 | 0 | 1 | 1 | 1 | 1 | 5 |
| Education/Outreach-Compliance | + | 1 | 1 | 1 | 0 | 1 | 0 | 1 | 5 |
| Invasive Species-Coral Reefs | - | 0 | 1 | 1 | 0 | 1 | 1 | 0 | 4 |
| Water Temperature-Coral Reefs | - | 0 | 1 | 1 | 1 | 0 | 1 | 0 | 4 |
| Non-Point Pollution-Water Quality | - | 1 | 1 | 1 | 0 | 1 | 0 | 0 | 4 |
| Coastal Development-Mangroves | - | 0 | 1 | 0 | 1 | 1 | 1 | 0 | 4 |
| Sargassum-Tourism | - | 1 | 1 | 1 | 0 | 0 | 1 | 0 | 4 |
| Commercial Fishers-Fisheries Resources | - | 1 | 1 | 1 | 0 | 0 | 1 | 0 | 4 |
| Sargassum-Water Quality | - | 1 | 0 | 1 | 1 | 1 | 0 | 0 | 4 |
| Marine Diseases-Fisheries Resources | - | 1 | 1 | 1 | 0 | 0 | 1 | 0 | 4 |
| Climate Change-Coral Reefs | - | 0 | 1 | 0 | 1 | 1 | 1 | 0 | 4 |
| Hurricanes-Coral Reefs | - | 0 | 1 | 1 | 0 | 1 | 1 | 0 | 4 |
| Law Enforcement-IUU Fishing | - | 0 | 0 | 1 | 1 | 1 | 1 | 0 | 4 |
| Water Temperature-Marine Diseases | + | 0 | 1 | 1 | 0 | 0 | 1 | 1 | 4 |
| Water Quality-Seagrass | + | 0 | 1 | 0 | 1 | 1 | 0 | 1 | 4 |
| Marine Habitat-Coral Reefs | + | 1 | 1 | 1 | 0 | 0 | 0 | 1 | 4 |
| Marine Habitat-Seagrass | + | 1 | 1 | 1 | 0 | 0 | 0 | 1 | 4 |
| Run Off-Pollution | + | 0 | 1 | 1 | 1 | 0 | 0 | 1 | 4 |
| Marine Habitat-Fisheries Resources | + | 1 | 1 | 1 | 0 | 0 | 1 | 0 | 4 |
| Fisheries Resources-Commercial Fishers | + | 1 | 1 | 1 | 0 | 0 | 1 | 0 | 4 |
| Climate Change-Marine Diseases | + | 1 | 0 | 1 | 1 | 1 | 0 | 0 | 4 |
| Law Enforcement-Management | + | 0 | 1 | 1 | 0 | 1 | 1 | 0 | 4 |
| Mangroves-Nursery Habitats | + | 0 | 1 | 1 | 1 | 1 | 0 | 0 | 4 |

Puerto Rico DAP (Groups 1 and 2) (See S2 File). The right side of the plot, where Businesses and NGOs are mostly found, is characterized by a concern over the impacts of marine diseases (e.g., stony coral tissue disease) on coral reef habitats (Group 4). The top portion of the plot, mostly driven by the St. Thomas/St. John DAP, is characterized by the importance of the relationship between management decisions and the development of fishery regulations. The bottom section of the plot, mainly influenced by the SSC conceptual model, is characterized by the impacts of anthropogenic forces, such as tourism, pollution and coastal development, on marine habitat including coral reefs, mangroves, and seagrass beds, as well as the importance of seagrass habitat for species of marine turtles. Stakeholder groups located in the middle of the plot (Group 3), which includes Academics and the St. Croix DAP, are those whose views represent a combination of the relationships explaining the distribution displayed in Fig 4 and for which no specific relationships emerge in the 2-dimensional space represented.

## 3.3. Priority ecosystem components

The diagrams displayed in Fig 5 show the collective perception of participant stakeholder groups regarding components that can be considered critical drivers and receivers of the U.S. Caribbean ecosystem, i.e., those which display high level of agreement between stakeholder groups. Components mentioned by all seven stakeholder groups (red circles in Fig 5) are listed

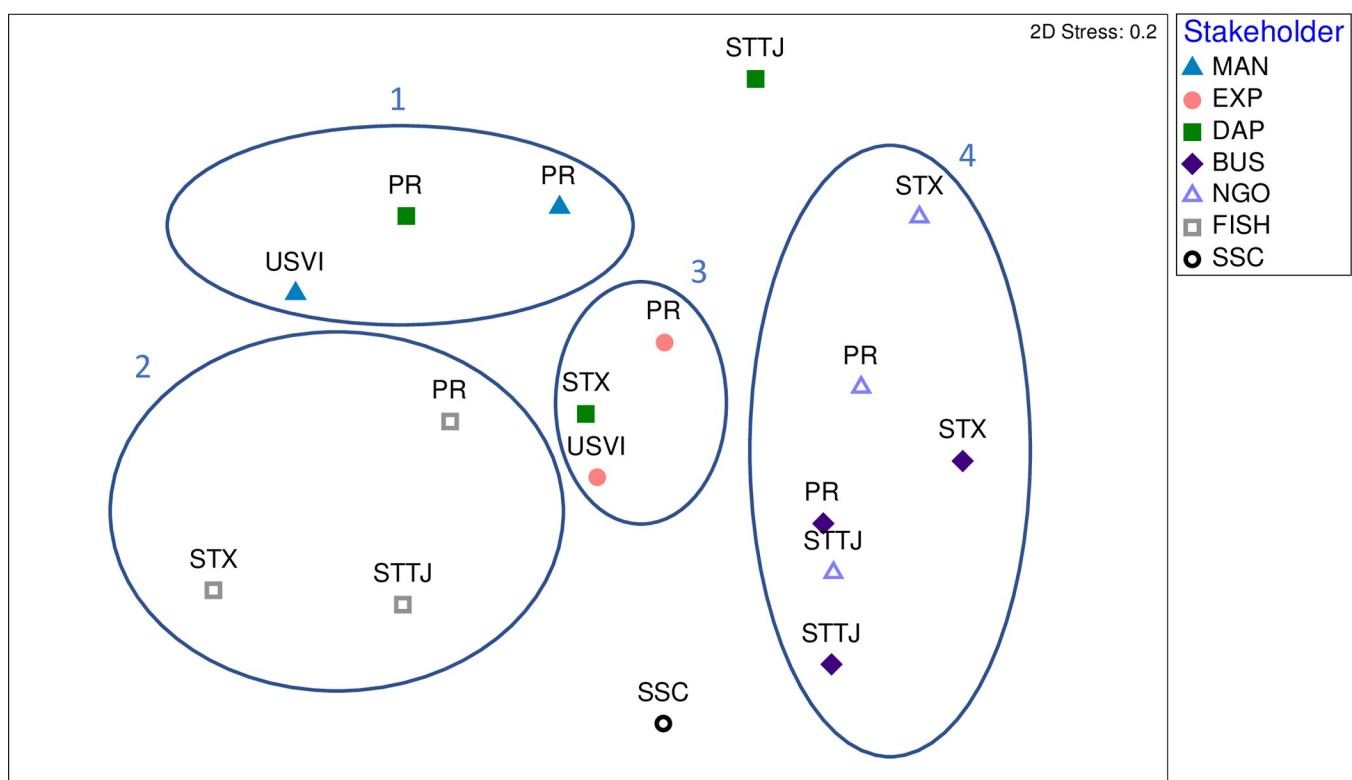

**Fig 4. nMDS ordination of stakeholder groups and U.S. Caribbean islands based on a simple matching coefficient between relationships of ecosystem components identified in conceptual models.** (MAN = managers; EXP = academics; DAP = District Advisory Panels; BUS = businesses; NGO = Environmental Non-Governmental Organizations; FISH = commercial fishers; SSC = Scientific and Statistical Committee; PR = Puerto Rico; STTJ = St. Thomas/St. John, STX = St. Croix; Ellipses = statistically significant groups defined by SIMPROF).

in Table 3. The diagrams displayed in Fig 5 also provide a visual representation of the number of unique components that were identified by each stakeholder group, representing the high diversity of perceptions captured in the melded model (Fig 3). Diagrams for each island/island group can be found in S3 File.

Table 3 identifies the components considered important elements of the U.S. Caribbean fishery ecosystem based on the maximum level of agreement between different stakeholder groups and can, therefore, be considered areas of common concern and priorities for stakeholders in the region. These include habitat integrity, water quality, the impacts of marine diseases, coastal development and pollution, the influence of recreational fisheries and tourism, as well as the positive impacts of education and outreach and local culture on the fishery ecosystem.

Ecosystem components that were characterized as major receivers based on stakeholder agreement level (Table 3), were selected and the most significant threats to them, i.e., elements that most stakeholders agreed had a detrimental impact on these receivers, were identified (Table 4). Run-off and marine diseases, while identified by stakeholders as important receivers (Table 3), were not included in Table 4 because when these components played the role of receivers, drivers maximize rather than impacted them in a detrimental way (e.g., *water temperature-marine diseases* or *river-run-off*). The major threats identified to marine habitat, water quality, recreational fishing activities, tourism, and outreach and education efforts included factors related to or compounded by climate change impacts, pollution, diseases, and anthropogenic pressures (Table 4).

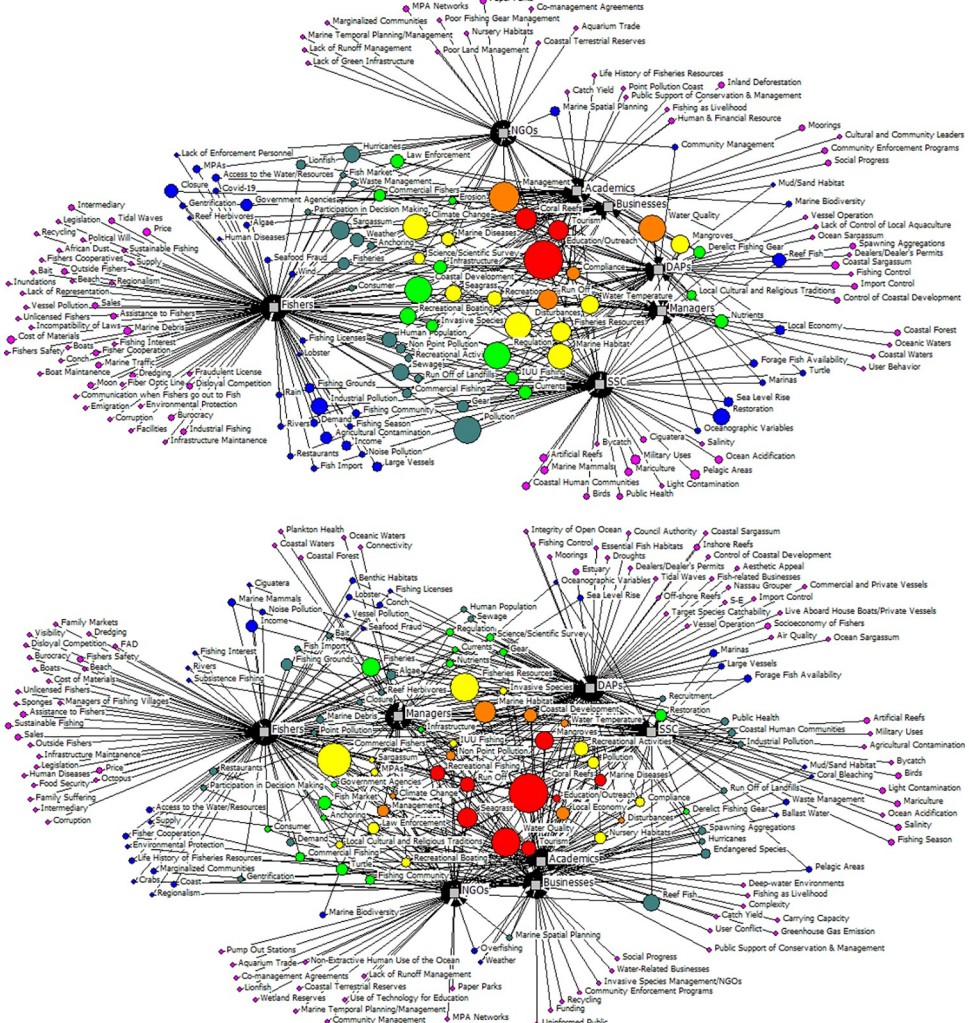

**Fig 5.** Two-mode network diagram showing all components identified as important drivers (A) or Receivers (B) of relationships (circles) by each stakeholder group (grey squares) in the U.S. Caribbean. The size of the circles is indicative of frequency with which component was mentioned by stakeholders. Circles color scheme indicates the number of stakeholder groups mentioning the component: red = 7; orange = 6; yellow = 5; light green = 4; dark green = 3; blue = 2; pink = 1.

## 4. Discussion

In this study, analyses of 29 stakeholder-driven conceptual models constructed using a fuzzy cognitive approach with over 230 participants helped to visualize and better understand the U. S. Caribbean fishery ecosystem from a stakeholder perspective and identify areas of concern among seven different fishery stakeholder groups. The large number of components and relationships found in the consensus model is indicative of the inherent diversity that characterizes the U.S. Caribbean region fisheries systems. The identification of ecosystem elements and threats considered collective priorities will be provided to federal and state agencies (e.g., CFMC, USVI Department of Planning and Natural Resources (DPNR) and Puerto Rico Department of Natural and Environmental Resources (DNER)) to advance fisheries management and policy in the region. These analyses emphasized the importance of participatory methods and co-production of knowledge to better understand factors affecting the region's

**Table 3. Components identified by all seven stakeholder groups as being important drivers (D) and/or receivers (R) in the U.S. Caribbean fishery ecosystem.**

| Ecosystem Components |
| --- |
| Recreational Fishing (D/R) |
| Coral Reefs (D/R) |
| Run-off (D/R) |
| Tourism (D/R) |
| Education/Outreach (D/R) |
| Marine Diseases (D/R) |
| Disturbances (D) |
| Coastal Development (D) |
| Local Cultural and Religious Traditions (D) |
| Water Quality (R) |
| Mangroves (R) |
| Seagrass (R) |

fisheries. In addition, the analyses identified several important ecosystem components that were unique to each stakeholder group. The development of separate conceptual models with different stakeholder groups, which increased the depth and diversity of data collected, expanded the analytical possibilities allowing for comparisons between different stakeholder groups to objectively understand convergence and divergence of perceptions and priorities.

One important outcome of the described efforts to develop conceptual models for the U.S. Caribbean was the schematic representation of the local fishery ecosystem as conceptualized

**Table 4. Major threats—characterized by the highest level of agreement between stakeholder groups—that affect the most significant receivers identified by all stakeholder groups.**

| Major Threats* | Components Affected |
| --- | --- |
| Run Off<br>Climate Change | Coral Reefs |
|  | Marine Habitat** |
|  | Water Quality |
| Hurricanes | Coral Reefs |
|  | Recreational Fishing |
| Tourism | Marine Habitat* |
|  | Education/Outreach |
| Non-Point Pollution | Marine Habitat* |
|  | Water Quality |
| Marine Diseases<br>Invasive Species<br>Water Temperature | Coral Reefs |
| Coastal Development<br>Disturbances<br>Anchoring | Marine Habitat* |
| Nutrients<br>Erosion<br>Point Pollution | Water Quality |
| Sargassum | Tourism |

*Identified by at least 3 stakeholder groups, except for Hurricanes-Recreational Fishing (2 group) and Tourism-Education/Outreach (1 group).

**Including mangroves and seagrass beds in addition to other habitats.

by diverse stakeholders. While conceptual models are widely used as tools to include and represent stakeholder perceptions in the conceptualization of fishery ecosystems [3, 12, 26, 27], this study is unique, to the best of the authors' knowledge, in its development of singular conceptual models with different fishery stakeholder groups. Analysis of similarity and agreement between stakeholder groups allowed for the identification of common concerns regarding important ecosystem components and their respective threats. Major ecosystem components characterized as receivers, i.e., affected by other components, include marine habitat, particularly coral reefs, water quality, recreational fishing activities, tourism, and education and outreach efforts. Stakeholders also recognized, through multiple relationships, the important interactions between different marine habitats, particularly positive feedback between coral reefs, mangroves, and seagrass beds. The most significant factors negatively affecting these elements, i.e., threats, include multiple sources of pollution, run-off, marine diseases, invasive species, coastal development, erosion, sargassum, tourism, and climate change impacts including water temperature changes, hurricanes, and coral bleaching, among other anthropogenic stressors. Many of these findings are in line with previous studies investigating U.S. Caribbean stakeholders', particularly fishers', perceptions of factors affecting the fishery and marine ecosystem, notably those related to pollution, habitat/coastal degradation, tourism, and climate change impacts [28–38]. Stakeholders also agreed on the positive influence of education and outreach efforts on compliance with rules and regulations and the effects of law enforcement in combating IUU fishing and contributing to fishery management objectives.

Analysis of similarity between different stakeholder groups among the different island/island groups showed significant cohesiveness among groups despite geographic location, suggesting that many of the identified concerns and priorities are shared within stakeholder groups throughout the U.S. Caribbean. The disparity observed among the DAPs can be potentially explained by the current heterogeneity in the composition of these committees among the three island/island groups. While the St. Thomas/St. John DAP is mostly comprised of commercial fishers, the St. Croix DAP includes members representing related businesses (e.g., professional divers), as well as academics and other fields. The Puerto Rico DAP has a majority of commercial fishers but includes more representation from recreational and aquarium trade sectors when compared to St. Thomas/St. John's. This might have contributed to the differing perspectives between these groups, but it does not explain the significant differences observed between the St. Thomas/St. John DAP and the commercial fishers. However, this observation emphasizes findings by [38], showing that many fishers in St. Thomas/St. John expressed discontent about adequate representation of their perspectives and priorities in the fisheries management process. More research is needed to further investigate these disparities and develop adequate responses to ensure a more participatory environment in fisheries management in the U.S. Caribbean. Despite these differences, analyses comparing stakeholder groups and geographic locations further demonstrate the importance of participatory approaches and the development of a shared vision and co-production of knowledge to address complex problems, as different groups clearly bring their unique set of perspectives to the table.

Conceptual modeling is well established as a tool to understand and explore behavior of complex systems, but its usefulness has shown limitation when outcomes are aimed at supporting decision making [21, 39]. This concern motivated, in part, the decision to develop separate conceptual models for different stakeholder groups, which allowed for more objective comparisons of similarities and differences between groups and minimized the effects of group dynamics. In this sense, the identification of areas of agreement between stakeholders regarding important ecosystem elements and their threats, not by achieving consensus through discussion but by the identification of similar elements and relationships among different conceptual models, provides a compelling method for prioritization to guide fisheries

policy and management actions. Furthermore, targeting areas and concerns that are common to multiple stakeholder groups can objectively promote collaboration and contribute to increased acceptance of and participation in the decision-making process [3, 11, 12, 15, 16].

The results of the conceptual model analysis presented here are currently being used in efforts by the CFMC to develop a Fishery Ecosystem Plan (FEP) for the U.S. Caribbean region as part of a transition to EBFM, exemplifying a successful use of conceptual model methodology to directly inform and influence management and policy decisions. The melded conceptual model for the region is being used to provide a conceptualization of the fishery ecosystem from the perspective of key stakeholders, and ecosystem elements characterized as priorities based on stakeholder agreement are being used to guide strategic objectives that will be used by the CFMC to guide future action. In addition, threats to ecosystem elements identified by stakeholders are also being used in the identification of risk factors as part of an effort to develop a risk assessment framework to guide management decisions by the CFMC and further advance EBFM in the region.

It is also noteworthy that relationships identified by stakeholders often involved components which are overseen by different government agencies (e.g., EPA and NOAA) and levels (e.g., federal v. local), thus action to address them would require interagency collaboration. The holistic approach proposed by EBFM, well reflected in the outcomes of the conceptual models, is often inconsistent with established governance structure, emphasizing the need for more effective collaborative strategies to address complex fishery management problems [39]. The identification of these "outside of jurisdiction" issues through the conceptual model process can, however, provide a tool for stakeholders to initiate and justify increased cooperation. Through the FEP process in the U.S. Caribbean, concerns and priorities identified in stakeholders' conceptual models which are outside of the scope of the CFMC are being used to provide a framework for future interagency cooperation using, for instance, letters and public support mechanisms to initiate action to address these issues.

Common concerns and priorities identified by stakeholders can also be used to identify data gaps and potentially guide the development of research priority documents and allocation of research funds. Addressing these knowledge gaps will help to advance management priorities and further understand interactions and cumulative impacts of factors outside of the jurisdiction of fisheries management agencies (e.g., pollution). Linking stakeholder concerns and research priorities also strengthens public participation in the policy process, providing opportunities for education and outreach efforts with a focus on increasing awareness and understanding of the decision-making process, and compliance with rules and regulations.

One important characteristic of conceptual models is their adaptable nature, supporting future iterations that can be used to strengthen, update, include other stakeholder groups (e.g. the U.S. Caribbean recreational sector) and mold models to include new perspectives and support adaptive management strategies. This will also help to further develop the communication language initiated here and refine methods for more effective interactions and consensus representation. Iterations of the conceptual model process can also be a useful tool to ensure continuous and increased stakeholder engagement, which can build upon initial momentum and lead to other fruitful collaborations. Further analysis of the data presented here, including quantitative and qualitative approaches to test hypotheses (e.g., testing relationships against existing biological, environmental, and socio-economic data), and scenario development, will help to inform and guide the continuation of this process and increase the usefulness of the conceptual modeling approach for fishery management purposes in the U.S. Caribbean region and elsewhere.

## Supporting information

**S1 Dataset.**
(XLSX)

**S1 File.**
(DOCX)

**S2 File.**
(DOCX)

**S3 File.**
(DOCX)

## Acknowledgments

The authors would like to acknowledge Braulio Quintero and Nicole Greaux for their leadership and assistance organizing and running stakeholder workshops, and the Lenfest Ocean Program team, especially Jason Landrum, Emily Knight, Kayla Ripple, and Victoria Bell, for assistance throughout project implementation and outreach.

## Author Contributions

**Conceptualization:** Tarsila Seara, Stacey M. Williams, Graciela Garcia-Molliner, Orian Tzadik, Michelle Duval, Juan J. Cruz-Motta.

**Data curation:** Tarsila Seara, Juan J. Cruz-Motta.

**Formal analysis:** Tarsila Seara, Kiara Acevedo, Juan J. Cruz-Motta.

**Funding acquisition:** Tarsila Seara, Stacey M. Williams, Juan J. Cruz-Motta.

**Investigation:** Tarsila Seara, Stacey M. Williams, Kiara Acevedo, Graciela Garcia-Molliner, Orian Tzadik, Michelle Duval, Juan J. Cruz-Motta.

**Methodology:** Tarsila Seara, Stacey M. Williams, Juan J. Cruz-Motta.

**Project administration:** Tarsila Seara, Stacey M. Williams, Juan J. Cruz-Motta.

**Supervision:** Tarsila Seara, Stacey M. Williams, Juan J. Cruz-Motta.

**Validation:** Tarsila Seara, Stacey M. Williams, Juan J. Cruz-Motta.

**Visualization:** Tarsila Seara, Kiara Acevedo, Juan J. Cruz-Motta.

**Writing – original draft:** Tarsila Seara, Stacey M. Williams, Juan J. Cruz-Motta.

**Writing – review & editing:** Graciela Garcia-Molliner, Orian Tzadik, Michelle Duval.

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
