## [Decision Letter · Decision Letter 0]

6 Feb 2024

PONE-D-23-38007Development and analyses of stakeholder driven conceptual models to support the implementation of Ecosystem-Based Fisheries Management in the U.S. CaribbeanPLOS ONE

Dear Dr. Ferreira Seara,

Thank you for submitting your manuscript to PLOS ONE. After careful consideration, we feel that it has merit but does not fully meet PLOS ONE’s publication criteria as it currently stands. Therefore, we invite you to submit a revised version of the manuscript that addresses the points raised during the review process.

**Two reviewers have assessed your manuscript and appreciate the value of your work, and how it can contribute to the larger academic community. They do however recommend some minor revisions to improve the readability of the manuscript. Besides their requests, I would also like to recommend improving the resolution of the figures. Some of them are hard to read or show pixelated on the submitted PDF. **

We look forward to receiving your revised manuscript.

Kind regards,

Juan Carlos Rocha Gordo

Academic Editor

PLOS ONE

Journal Requirements:

3. We note that your Data Availability Statement is currently as follows: All XXX files are available from the XXX database (URL XXX)

Reviewers' comments:

Reviewer's Responses to Questions

**Comments to the Author**

1. Is the manuscript technically sound, and do the data support the conclusions?

Reviewer #1: Yes

Reviewer #2: Yes

2. Has the statistical analysis been performed appropriately and rigorously? 

Reviewer #1: Yes

Reviewer #2: Yes

3. Have the authors made all data underlying the findings in their manuscript fully available?

Reviewer #1: No

Reviewer #2: No

4. Is the manuscript presented in an intelligible fashion and written in standard English?

Reviewer #1: Yes

Reviewer #2: Yes

5. Review Comments to the Author

Reviewer #1: I believe that the manuscript is clearly written and with sufficient explanation of the analyses to be understandable, but suitable for a general audience less familiar with multivariate methods, network analysis and fuzzy logic models. In addition, the context given in the introduction section makes it clear to understand the relevance of the study and the findings.

One of the things I might suggest is that the discussion section lacks some ideas on the state of the art regarding the implementation of the ebfm approach in data-poor situations in other regions of the world. As well as methodological advances in this topic.

Some particular comments are listed below:

Abstract. Some ideas about used methodology for analyses would be recommendable.

Lines 117-118 It is not clear how or which efforts were made.

Table 1. Please define acronyms.

Lines 157. How was the system delimited? Where it starts or ends? I think this is relevant to clarify what kind of components could be named by participants.

Lines 164-167. Are there moderators at online workshops/platforms? What is the profile of moderators?

Lines 216-239. Some information about figures is repeated in the text. Please avoid this in all results section.

Table 2. Explain in Table title that # of groups are the number of groups that identify the same relationship (or something similar).

Fig 4 and Figs 5. Needs a higher resolution.

Table 3. Wouldn't it be useful to also consider other components as important if they were mentioned by the majority? For example, let's say more than four groups. Why only those where everyone agreed are considered important. Perhaps it would be worthwhile to separate between priority (total consensus) and relevant (mentioned by several groups)?

Line 345. Can you discuss what are benefits of developing several models from different groups vs a unique model? Also, you end with a consensus model, it is not the same that building a unique model like in other studies? Can you bring some ideas to discussion? For example, what are the lessons learned for stakeholders when comparing their different perspectives?

Reviewer #2: The paper documents the process of developing and analysing stakeholder-driven conceptual models with seven different stakeholder groups in Puerto Rico and the US Virgin Islands to assess and compare their perceptions of the fishery ecosystem so as to support the implementation of a Fishery Ecosystem Plan (a document to guide efforts in adopting Ecosystem Based Fisheries Management). I found the paper to be well written, informative and novel and feel that it can provide a useful guide for other agencies and practitioners around the world seeking to implement a similar process. I feel that it should be accepted for publication in PLOSOne pending some minor revisions which are outlined below.

1) I suggest making the Abstract shorter by limiting the background information presented between lines 23-28 and focusing on what the study did.

2) At line 45, the first mention of the U.S. should be written as "United States (U.S.).

3) At line 57, you mention that efforts to adopt an ecosystem approach to fisheries management in the U.S Caribbean were initiated in 2012. It would be nice to give readers a sense of how much later this was compared to the broader U.S.

4) The use of "knowledge co-production" at the end of lines 84-85 feels a little awkward and confusing to me. Maybe try some thing like "shared knowledge acquisition and exchange".

5) At line 86, you mention traditional and non-traditional experts. In this context are you talking about traditional (i.e.: Indigenous) people and non-traditional people? Or are you referring to those historically viewed as experts (i.e.: scientists and managers) versus non-experts?

6) At line 88, please change "are associated" to "is associated".

7) At line 90, change "transitioning" to "transition".

8) You mention interactions with recreational fisheries and tourism at line 98. Were stakeholders from the recreational (including Charter tourism) sector and artisanal/Indigenous sectors included (and explicitly differentiated from commercial fisheries in terms of their unique characteristics, priorities and perceptions of the fishery ecosystem) in the process of developing the conceptual models? If not, I think it is worth either acknowledging this as a limitation of the study or justifying their omission in the process. The definition of EBFM at lines 45-51 implies that all affected fishery components and thus, presumably, all sectors responsible for these pressures should be considered. If these groups were included; they need to be to be explicitly highlighted in Table 1.

9) At line 152, since the reference is actually part of the sentence here it should be written as "Ozesmi & Ozesmi [19]".

10) At line 224, please replace "are priorities" with "as priorities".

11) Lines 440-441, please populate acknowledgment text or omit section.

6. PLOS authors have the option to publish the peer review history of their article (what does this mean?). If published, this will include your full peer review and any attached files.

Reviewer #1: No

Reviewer #2: **Yes**

---

## [Author Response · Author response to Decision Letter 0]

2 Apr 2024

Reviewer #1: I believe that the manuscript is clearly written and with sufficient explanation of the analyses to be understandable, but suitable for a general audience less familiar with multivariate methods, network analysis and fuzzy logic models. In addition, the context given in the introduction section makes it clear to understand the relevance of the study and the findings.

One of the things I might suggest is that the discussion section lacks some ideas on the state of the art regarding the implementation of the ebfm approach in data-poor situations in other regions of the world. As well as methodological advances in this topic.

- While this is an important topic and the authors appreciate the suggestion, it was not a major objective of this particular paper to provide a worldwide comparison nor to review the different methods currently employed. This manuscript’s goal is to share results obtained for the US Caribbean and discuss the importance of the findings in the context of policy and management for the region. 

Some particular comments are listed below:

Abstract. Some ideas about used methodology for analyses would be recommendable.

- Added info on methods to abstract.

Lines 117-118 It is not clear how or which efforts were made.

- Edited to increase clarity. 

Table 1. Please define acronyms.

- Location/island acronyms added below table; other acronyms have been defined in the text. 

Lines 157. How was the system delimited? Where it starts or ends? I think this is relevant to clarify what kind of components could be named by participants.

- Edited to increase clarity: Participants were first asked to identify important components of the entire U.S. Caribbean marine fishery system, which could be social, biological, economic, cultural, or physical in nature, and then prompted to link these different components based on relationships between them.

Lines 164-167. Are there moderators at online workshops/platforms? What is the profile of moderators?

- Edited: During online workshops, models were constructed by the research team on the screen using Mental Modeler online software

Lines 216-239. Some information about figures is repeated in the text. Please avoid this in all results section.

- The figure captions are included next to the text close to where figures are mentioned. The information in the text is needed for context and the authors believe that removing that information from the text will negatively impact the reader’s comprehension of the content. We would appreciate further guidance from the editorial team. 

Table 2. Explain in Table title that # of groups are the number of groups that identify the same relationship (or something similar).

- Edited: Table 2. List of top relationships by stakeholder agreement level and respective stakeholder groups for the U.S. Caribbean. Number of groups refer to the total number of stakeholder groups that identified each relationship in their conceptual model(s).

Fig 4 and Figs 5. Needs a higher resolution.

- We have provided files with improved resolution for these 2 figures. 

Table 3. Wouldn't it be useful to also consider other components as important if they were mentioned by the majority? For example, let's say more than four groups. Why only those where everyone agreed are considered important. Perhaps it would be worthwhile to separate between priority (total consensus) and relevant (mentioned by several groups)?

- The authors acknowledge that the data collected can be analyzed in myriad different ways to highlight different important aspects of stakeholder perceptions and priorities with regard to the ecosystem. While we appreciate the reviewer’s comment, we believe that we have provided enough examples and covered the idea of relevancy and consensus extensively in the text – Figure 5 is an example of that. Aspects of consensus v. relevancy are also emphasized in the discussion as well as in other analyses in the paper. The authors disagree that the inclusion of Table 3 disregards the importance of agreement without total consensus. 

Line 345. Can you discuss what are benefits of developing several models from different groups vs a unique model? Also, you end with a consensus model, it is not the same that building a unique model like in other studies? Can you bring some ideas to discussion? For example, what are the lessons learned for stakeholders when comparing their different perspectives?

- The authors believe that this topic is adequately covered later in the discussion, specifically, in the following paragraph: Conceptual modeling is well established as a tool to understand and explore behavior of complex systems, but its usefulness has shown limitation when outcomes are aimed at supporting decision making [21, 39]. This concern motivated, in part, the decision to develop separate conceptual models for different stakeholder groups, which allowed for more objective comparisons of similarities and differences between groups and minimized the effects of group dynamics. In this sense, the identification of areas of agreement between stakeholders regarding important ecosystem elements and their threats, not by achieving consensus through discussion but by the identification of similar elements and relationships among different conceptual models, provides a compelling method for prioritization to guide fisheries policy and management actions. Furthermore, targeting areas and concerns that are common to multiple stakeholder groups can objectively promote collaboration and contribute to increased acceptance of and participation in the decision-making process [15, 16, 11, 12, 3]. 

Reviewer #2: The paper documents the process of developing and analysing stakeholder-driven conceptual models with seven different stakeholder groups in Puerto Rico and the US Virgin Islands to assess and compare their perceptions of the fishery ecosystem so as to support the implementation of a Fishery Ecosystem Plan (a document to guide efforts in adopting Ecosystem Based Fisheries Management). I found the paper to be well written, informative and novel and feel that it can provide a useful guide for other agencies and practitioners around the world seeking to implement a similar process. I feel that it should be accepted for publication in PLOSOne pending some minor revisions which are outlined below.

1) I suggest making the Abstract shorter by limiting the background information presented between lines 23-28 and focusing on what the study did.

- Done

2) At line 45, the first mention of the U.S. should be written as "United States (U.S.).

- Done

3) At line 57, you mention that efforts to adopt an ecosystem approach to fisheries management in the U.S Caribbean were initiated in 2012. It would be nice to give readers a sense of how much later this was compared to the broader U.S.

- This is difficult to answer since the different regions have been at various stages of developing and implementing EBFM strategies for many decades and the goals and methods are varied and region specific. This comparison would likely be lengthy and not in line with the objectives of the paper. 

4) The use of "knowledge co-production" at the end of lines 84-85 feels a little awkward and confusing to me. Maybe try some thing like "shared knowledge acquisition and exchange".

- We believe the term co-production of knowledge is well established in the literature and it is being correctly employed here. 

5) At line 86, you mention traditional and non-traditional experts. In this context are you talking about traditional (i.e.: Indigenous) people and non-traditional people? Or are you referring to those historically viewed as experts (i.e.: scientists and managers) versus non-experts?

- The former is correct – examples were added to the text to increase clarity: Engaging diverse stakeholders offers several advantages, including bridging the gap between traditional (e.g., scientists) and non-traditional experts (e.g. fishers),

6) At line 88, please change "are associated" to "is associated".

- This sentence has been edited for clarity. 

7) At line 90, change "transitioning" to "transition".

- Done.

8) You mention interactions with recreational fisheries and tourism at line 98. Were stakeholders from the recreational (including Charter tourism) sector and artisanal/Indigenous sectors included (and explicitly differentiated from commercial fisheries in terms of their unique characteristics, priorities and perceptions of the fishery ecosystem) in the process of developing the conceptual models? If not, I think it is worth either acknowledging this as a limitation of the study or justifying their omission in the process. The definition of EBFM at lines 45-51 implies that all affected fishery components and thus, presumably, all sectors responsible for these pressures should be considered. If these groups were included; they need to be to be explicitly highlighted in Table 1.

- We have added a definition for “commercial fishers” under table 1 to make it explicit that our models did not include the recreational sector. The reason for not including them was logistical (and certainly a weakness) but the reality is that we did not include every possible fishery stakeholder in our conceptual models. The importance of iteration is stressed in the discussion and future conceptual models in the region should include the recreational sector – we have added a line emphasizing that in the last paragraph of the discussion. 

9) At line 152, since the reference is actually part of the sentence here it should be written as "Ozesmi & Ozesmi [19]".

- Done

10) At line 224, please replace "are priorities" with "as priorities".

- Done

11) Lines 440-441, please populate acknowledgment text or omit section.

- Acknowledgements have been added.

---

## [Editor Report · Decision Letter 1]

7 May 2024

Development and analyses of stakeholder driven conceptual models to support the implementation of Ecosystem-Based Fisheries Management in the U.S. Caribbean

PONE-D-23-38007R1

Dear Dr.Ferreira Seara,

We’re pleased to inform you that your manuscript has been judged scientifically suitable for publication and will be formally accepted for publication once it meets all outstanding technical requirements.

Kind regards,

Juan Carlos Rocha Gordo

Academic Editor

PLOS ONE

---

## [Editor Report · Acceptance letter]

17 May 2024

PONE-D-23-38007R1 

PLOS ONE

Dear Dr. Ferreira Seara, 

I'm pleased to inform you that your manuscript has been deemed suitable for publication in PLOS ONE. Congratulations! Your manuscript is now being handed over to our production team.

Kind regards, 

on behalf of

Dr. Juan Carlos Rocha Gordo 

Academic Editor

PLOS ONE